# A Simulation Study on Spread of Disease and Control Measures in Closed Population Using ABM

Youngmin Kim 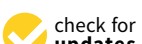 and Namsuk Cho *

Department of Operations Research, Korea National Defence University (KNDU), Nonsan 33021, Korea; youngmin4507@naver.com
* Correspondence: ncho64@gmail.com

**Abstract:** An infectious disease can cause a detrimental effect on national security. A group such as the military called a "closed population", which is a subset of the general population but has many distinct characteristics, must survive even in the event of a pandemic. Hence, it requires its own distinct solution during a pandemic. In this study, we investigate a simulation analysis for implementing an agent-based model that reflects the characteristics of agents and the environment in a closed population and finds effective control measures for making the closed population functional in the course of disease spreading.

**Keywords:** disease spread model; agent-based model; military and security; closed population

## 1. Introduction

The spread of infectious diseases has a great impact on human society. COVID-19 is the best representative disease that is currently impacting human society. It is an infectious disease that first occurred in Wuhan, China in 2019 [1]. Until now (2021), it has spread all over the world, affecting various areas of human society [2]. In most countries, the situation has changed a lot before and after COVID-19. Due to the government's disease control, contact between people has been minimized, and many industrial structures have been changed. Examples include the fields of culture, the economy, and social life. In particular, the spread of a large-scale disease can have a fatal impact on national security [3]. The impact of disease on national security can be explained in two ways.

First, it could threaten the defense readiness condition (DEFCON). Most countries have a particular level of DEFCON that must be secured in any situation, to a greater or lesser extent. All operation schemes during wartime are elaborately planned and trained under the assumption that such a level is strictly secured. When the disease spreads to the military, it obviously becomes an unpredictable event and it could be a fatal problem. For instance, in July 2021, a large-scale spread of COVID-19 occurred at the Cheonghae Naval unit (it is the Republic of Korea Navy Somali Sea Escort Task Group). It infected 90% of the total population within 20 days and showed the dangers of disease spread within the military. Moreover, DEFCON is not calculated by just the sum of available troops but by an evaluation of the complex functionality of the military. The military is an independent institution with various functions such as logistics, financial, intelligence, etc. It follows that complete paralysis of any part of the military can lead to the absence of function of the entire military. That could happen as the disease spreads in a specific group.

Second, the absence of the military function can cause a nationwide catastrophe extending to all of society. In most countries, especially during the early phase of the pandemic, the military provides urgent aid to society, including but not limited to emergency care in military medical facilities, transportation of vaccines, and the execution of declaring martial law in the worst-case scenario. Therefore, as 'first in and last out' resources of the nation, the military requires special attention as a preventative force against disease spreading.

For this reason, we believe that disease is an *invisible threat* and it is probable that one can deliberately conspire to weaponize diseases, as many other researchers mentioned [4].

Organizations such as the military that guarantee national security have different characteristics from the general society. Such a group can be called a "closed population" (CP). It is simply a subset of the general population (GP), but it has two distinct characteristics. First, CP has limited contact from outside. For example, the military has barracks that separate soldiers from civilians. In fact, there are members of CP residing outside of CP as well as civilian employees who can come and go frequently. However, such a group is a small part of CP. Second, members of CP are usually well-disciplined and can be controlled easily since the military has a hierarchical organizational structure and strong leadership. Even if we focus on the military, there are other kinds of groups that can be referred to as CP, such as religious facilities, research facilities, etc. We emphasize the fact that CP has distinct characteristics from GP and it will have a different pattern or aspect with respect to the spread of disease and control of disease spreading.

In this study, we use an agent-based model (ABM) to describe the dynamics of infectious disease spreading in CP. Through this model, we analyze the extent of the damage and disease spreading time. In addition, we introduce various control measures to block the spread of disease in CP and discuss the effectiveness of such measures with various simulation experiments. Before going into further details, we note that we use COVID-19 data for empirical research; however, this model is not focused only on COVID-19. The disease spread model in this study can be extended to any infectious disease.

## 2. Literature Review

There are various models that explain the mechanisms of infectious disease [5]. Many researchers study to determine the characteristics and scale of infectious diseases using disease spread models. Results using such models can be used to establish policies against disease. As diseases continue to evolve and change, researchers attempt to construct robust and accurate models. In general, disease spread models can be divided into mathematical models and simulation models.

### 2.1. Mathematical Model

One of the most popular disease models is the SIR model [6]. The philosophy of the SIR model is to be able to observe the disease from a macro perspective with some information related to the disease [7–9].

- "*S*" means "susceptible group". Susceptible groups can be infected with contact from an infected group.
- "*I*" means "infected group". The infected group is a state of being infected with the disease. Thus, the disease can be transmitted to a susceptible group.
- "*R*" means "recovery group". The recovery group is in a state of recovery from disease.

In the SIR model, each state is defined by the following simple equations:

$$dS/dt = -\beta SI \tag{1}$$

$$dI/dt = \beta SI - \gamma I \tag{2}$$

$$dR/dt = \gamma I \tag{3}$$

Changing the number of *S* to the number of *I* depends on the infection rate ($\beta$) and changing the number of I to the number of *R* depends on the recovery rate ($\gamma$). In addition, the basic infection reproduction number ($R_0$) can be estimated using infection rate ($\beta$) and recovery rate ($\gamma$), which is very important information for disease control [10].

$$R_0 = \tau * N_0 * d \tag{4}$$

$R_0$ is calculated using Equation (4). In Equation (4), $\tau$ is the transmissibility of the disease. $N_0$ is the number of a contact. d is the duration of the infectious period. In this equation, $N_0$ is the parameter related to human behavior, and the rest are related to the disease. So, $R_0$ is affected by the number of contacts and it may vary depending on the region. $R_0$ is the number of infections from one infected person to another person. If $R_0 > 1$, it is considered as a pandemic situation and if $R_0 < 1$, it implies that degree of disease spreading is becoming smaller. This value does not change. So people either use the $R_E$ value or use the $R_t$ value to compare it to the $R_0$ value [11]. $R_E$ is the average number of new infections transmitted by one infected person in an already infected population. $R_t$ is the average number of new infections that one infected person transmits over a certain period of time. $R_t$ and $R_E$ may be collectively called the infection reproduction number $R$, which should be managed separately from $R_0$. Therefore, researchers try to make $R_0 > R$, and ultimately aim to lower it to a value less than 1 [12]. These mathematical models are still useful for predicting disease spread. In addition, by improving the accuracy of the SIR model, it provides good insight for policy evaluation [13]. However, even though results from the SIR model are based on exact mathematical formulation, the results are quite different from those in the real world, and the model requires continued evolution.

Extending the SIR model involves two key ideas. The first idea is to subdivide the class [14]. In Figure 1, there are the basic SIR model and its variations. To improve the model, it is necessary to understand the practical characteristics of the disease. For example, empirically, people have a certain incubation period before they show symptoms of the disease. Such an observation leads to adding the exposed class to the model, and the model becomes an SEIR model. Similarly, if we consider some immune people, SIR has evolved into an MSEIR model. Lastly, if we consider life cycles such as the birth and the death [15] of the population, it becomes an extended model of MSEIR as we can see in Figure 1d.

The second approach to improving the SIR model is to use more accurate parameters. In mathematical models, parameters are critical for representing disease flows [16]. The researchers use critical parameters (e.g., $\beta$, $\gamma$) from real data obtained from similar diseases or estimate parameters by minimizing the sum of the squared deviations between the actual observed value and the hypothesized expected value [17]. However, since it is difficult to obtain real data, especially in the early phase of the disease, people use stochastic methods [18,19]. Even though stochastic transmission conditions include random noise, it is known that such stochastic models explain disease spreading well. Nonetheless, there is a limit to mathematical models in which they do not fully account for the complexity of the disease, and it is difficult to perform sensitivity analysis, which is required to induce applicable policy.

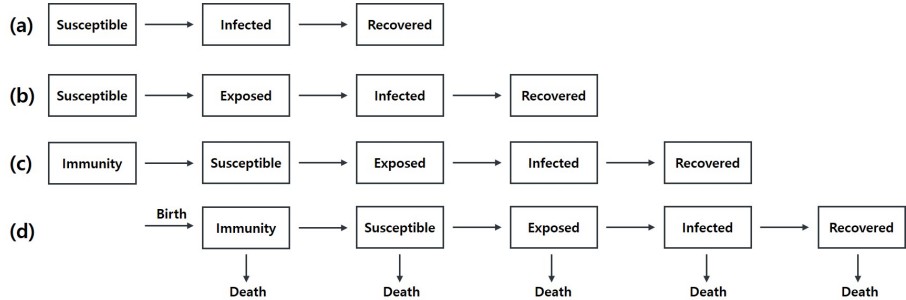

**Figure 1.** Examples of SIR model: (**a**) basic model; (**b**) SEIR model; (**c**) MSEIR model; (**d**) add life cycle parameter (death, birth), MSEIR model.

## 2.2. Simulation Model

An ABM is a computational model that simulates complex phenomena through interactions between agents. Most of the phenomena are of various kinds, and the pattern of occurrence is non-deterministic. Using ABM, complex phenomena can be explained with a few representative rule [20]. Therefore, by using the characteristics of ABM, various

complex environments can be expressed. For example, it is widely used in the fields of the stock market, supply chain network, transportation network, detection and search in military applications, etc. The disease is also a complex system, and studies have been conducted on a model to which special rules (i.e., disease transmission, human behavior, etc.) are applied using ABM.

A simple but useful disease spread model using ABM is shown in Figure 2. These models only require the input of three parameters: population number, infection rate, and recovery rate. In this model, susceptible agents are created as many as population numbers. Susceptible agents can be infected with respect to infection rate. Lastly, infected agents are recovered according to their recovery rate. This process is similar to the mechanism of the SIR model. However, unlike the SIR model, it is possible to visually observe the actions of the agents and various intuitive analyses are possible with experimental data. Further, the complexity of human society is reflected through randomness. Even if such models explain disease spreading broadly, they may miss a critical characteristic of the spread of the disease. For example, contact is one of the important factors in disease spread. In order to reflect the contact factor in the model, it is necessary to further define additional parameters such as population density, speed of people, or size of the space (facility). Another important factor is group characteristics. For instance, agents belonging to hospitals, schools, or the military may behave differently. In some studies, a model is created by subdividing the agent's individual properties. For example, agents can be classified by age or job, and different activity levels are applied [21] and it also find the best control measures in a variety of environments [22].

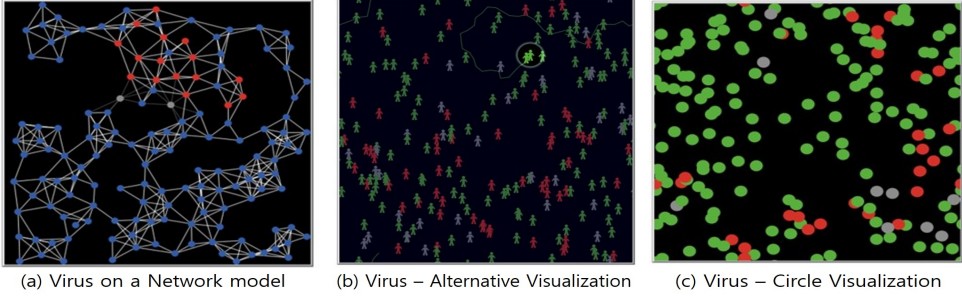

(a) Virus on a Network model      (b) Virus − Alternative Visualization      (c) Virus − Circle Visualization

**Figure 2.** Examples of disease models using ABM [23]: (**a**) spread of viruses on computer networks; (**b**,**c**) this model visualized the spread of disease in human society.

In this section, we reviewed two methodologies for describing the spread of a disease. The contribution of this work is twofold.

- First, our work is the first research that deals with disease spreading and its effect on CP. Previous studies mostly account for disease spreading in GP [24–27].
- Second, we focus on understanding the effects of various control measures that prevent the spread of infectious disease in CP. As we described in Section 1, comprehension of the effectiveness of control measures and finding the best strategy is important.

This paper is organized as follows: In Section 3, we describe the agent's properties, behaviors, and rules of control measures. In Section 4, we provide the results of the models and associated findings. In Section 5, we define human behaviors that affect disease, incorporate them into our models, and apply controls to measure them. Finally, in Section 6, conclusions are drawn.

## 3. Methodology

In this section, we define agent properties and explain behavior rules to express the characteristics of various populations. It also includes the principles of disease transmission rules and control measures. The assumptions of this study are as follows:

- The experiment is conducted in a limited space in order to control the population density of the GP. For the ratio of population density required for defining the population, demographic information from the Republic of Korea is used. (e.g., Seoul, Metrocity (Daejeon), Country (Nonsan)) [28].
- The data, such as the rate of asymptomatic infections, are obtained from the official data of the Ministry of Health of the Republic of Korea [29,30].
- This model ignores life cycles such as birth, death, and migration.

Based on these assumptions, this section explains the definitions of agent, environment, behavior, and controls measures used in modeling.

### 3.1. The Agent and Environment

In this model, an agent is a person in each population. As shown in Figure 3, agents are divided into two groups: those in CP and those in GP.

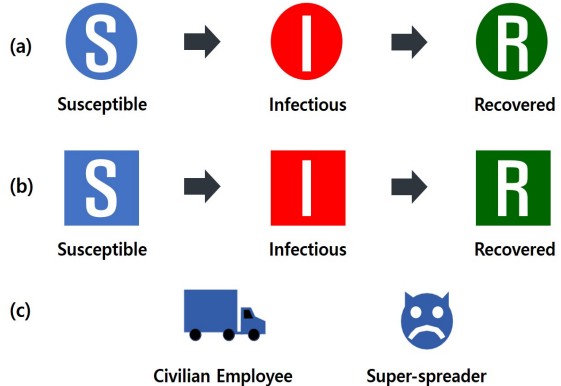

**Figure 3.** Symbol of agents of the model: (**a**) agents in a GP; (**b**) agents in a CP; (**c**) special agents (Civilian employee, Super-spreader).

In GP, agents are free to act outside. However, this agent cannot enter a CP. With this, the first characteristic, "limit contact from outside", is reflected in the model. In addition, the CP agent can only go outside with a certain ratio of agents. In this model, 30% of the population is free to go outside. Agents who go out are not subject to the rules of the CP and behave similarly to GP agents; however, after a certain period of time (e.g., vacation time), these agents must come back inside. CP agents must follow various controls. However, these controls apply when the agent is inside and not when the agent goes outside. In this way, based on the boundary of the CP, the strong control of the agent is expressed as the second characteristic, "well-disciplined".

An exceptional agent is needed, as shown in Figure 3c. First, it is a civilian worker in the military. They work inside the military only during office hours, and they move outside the rest of the time. Unlike CP agents, they have no obligation to return. When they are inside, they must follow the same rules as agents in CP.

The second one is a Super-spreader [31]. This is one of the important factors influencing the spread of disease. Super-spreaders cause more than twice as many infected people as general infected people [32]—so, it causes a lot of positive tested cases in a short time. In South Korea, there is a case where a large-scale positive test case was caused by a Super-spreader. As it appeared, Korea went through a pandemic phase and made many sacrifices. So, in this study, the agent properties are defined in order to see the Super-spreader's spread effect and block of control measure in action.

Figure 4 is a part world of the simulation model. To conduct an experiment reflecting two distinct worlds, a black border is used to divide the space. So, the external and internal populations can be adjusted according to user definition. The pink patch is the area of GP and the yellow patch is the area of CP. The military gate is the only passage that connects the outside and the inside. In this model, the control measures are applied only to the CP.

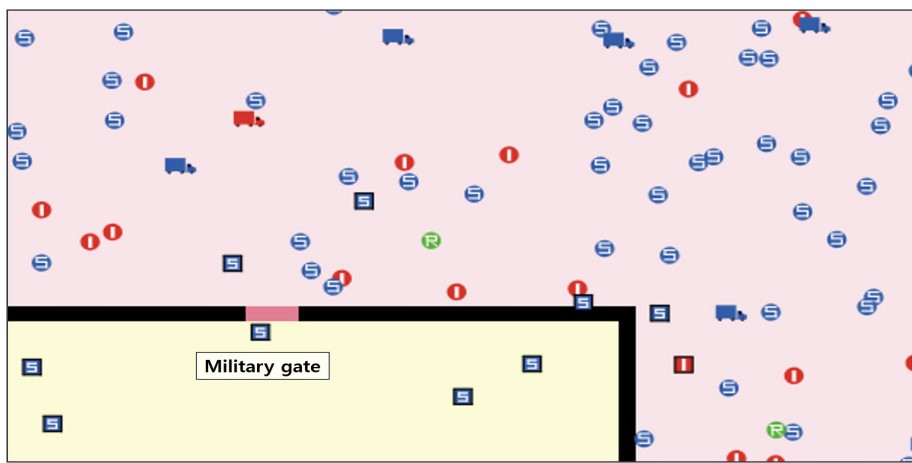

**Figure 4.** The part of the simulation environment.

*3.2. The Agent Behavior*

A useful model should be able to explain a variety of complex phenomena with a few simple rules. In this model, we use only three simple rules to describe disease spreading. Before we explain such rules, we define sets as follows:

[Set definition]

$A(= A_{IN} \cup A_{OUT})$ : A set of agents in population

$A_{IN}$ : A set of agents in Closed Population.

$A_{OUT}$ : A set of agents in General Population.

T : A set of time domain.

The first rule is the move of agents. Simply, the agent moves randomly left and right, forward and backward. The exact definition of agent movement is as follows:

$$\text{Let } x_t^i \in \mathbb{R}^2 \text{ be location of agent } i_{\in A} \text{ at time } t_{\in T}. \text{ then } x_{t+1}^i = x_t^i + \alpha d, \qquad (5)$$

$$\text{where } d \text{ is two dimensional vector such that angle between}$$

$$x_t^i \text{ and } d \text{ is less than } \frac{\pi}{6} \text{ and } \alpha \text{ is step length.}$$

$$(\tfrac{\pi}{6} \text{ is a half of human sight range.})$$

The second one is the transmission rule. Agents move erratically and may come into contact with infected agents. Then, the disease spreads from the infected agent and becomes a secondary infection. However, all contacts do not imply infections. Some individuals may not be infected with diseases due to their excellent immune function or high level of hygiene. To reflect this stochastic situation, probability is reflected in the transmission rule.

$$\text{Agent } i \text{ turns into infectious agent if,}$$

$$\text{There exist infectious agent } j \text{ such that } \|x_t^j - x_t^i\| \leq 2, \text{ and}$$

$$\text{Random number } \delta \in [0,1] \leq \text{Transmission rate} \qquad (6)$$

As in rule (6) (Transmission rate is based on research data [33]), an agent is infected when both two independent conditions are satisfied. When a disease spreads, it has a fixed transmission rate, and a random number is generated for each agent and compared with the transmission rate. Random number follows uniform distribution in [0, 1].

The third rule is recovery. It is the rule by which infected agents are recovered. Infected people recover after a certain period time. But, this time varies depending on the case. So, the recovery time applied to each individual is expressed as a probability.

$$Infected\ agent\ i\ turns\ into\ recovered\ agent\ if,$$

$$Random\ number\ \delta \in [0,1] \leq (1/recoveryday) \tag{7}$$

(The recovery date is 14 days, the maximum recovery period for COVID-19 [34]).

### 3.3. The Control Measures

In this study, we describe four control measures currently being applied to the South Korean military. The applied control measures are as follows:

The first measure is No-control. It is a state in which uncontrol is applied to all agents. Through this, we can experiment with the rate and dynamics of the spread of disease in an uncontrolled situation. The second measure is social distance. It is a control rule to limit the amount of person-to-person contact. In Korea, quarantine rules such as the prohibition of gatherings of more than five people are used. In this model, agents are slowed down to express minimal contact between people. If the moving speed of the agents is slowed, the number of contacts during the same period also decreases. The third one is screening. It is to control the number of people entering and leaving the military gate. The military gate checks the temperature of the influx of outside personnel and allows only low-temperature personnel to enter. However, sometimes an infected person can come inside. This is because there are infected people without fever symptoms, such as asymptomatic infected people. So, we applied the proportion of asymptomatic to the screening accuracy. According to the Korean Ministry of Health, the asymptomatic rate is about 33%. Therefore, the screening in this model blocks external infections with a 66% efficiency. In addition, people with fever symptoms should have their infection status reconfirmed through a PCR test. The fourth measure is the PCR test. The PCR test is a biologically validated and reliable control measure. Therefore, the accuracy is high (99%) when checking whether an external person is infected. Infected people are isolated right away if their infection is confirmed by a PCR test. So, it is a necessary measure when an agent is exposed to the outside for a long time or when unauthorized personnel enters.

We present a formal representation of the control measure in Appendix A.

## 4. Computation Result: Control Measures

In this section, we present the experimental results for the four control measures. Fundamentally, the measure of effectiveness is the dynamics of the number of infected people with respect to time. In order to analyze the sensitivity of the control measures, in each experiment, only one control method is applied where the others are fixed. In addition, to observe the effect of changes in external population density on CP, the population of the CP is fixed and the GP density is based on demographics data.

### 4.1. Validation of the Model

It is difficult to validate the simulation model [35]. We chose a practical validation method in the field in which we compare our results with the well-known SIR model.

Figure 5 is a graph of comparison between the mathematical model and the simulation model. The same $R_0$ value is applied to each model, and the trend line of the graph is compared. The graph in the mathematical model seems to be deterministic. However, we can see that the ABM model is slightly different from the mathematical model. The ABM model shows the stochastic result similarly to Figure 5c. So, 100 replications were carried out and the average value of the experimental results are expressed as a graph in Figure 5b. In addition, it is confirmed that there is no skewedness of data through comparison of average and median value. At this time, it is confirmed that Figure 5a,b are expressed as a graph with a similar trend. In addition, we check the similarity of the two graphs through time scaling. In Figure 5d, it can be seen that the ABM model overestimates the number of infected people more than the mathematical model. Further, there seems to be a difference in the slope and curve of the graph are. It seems to indicate that the random behavior of agents is well reflected in our ABM model. Although the two models are not perfect matches, their peak points and the graph trend are similar.

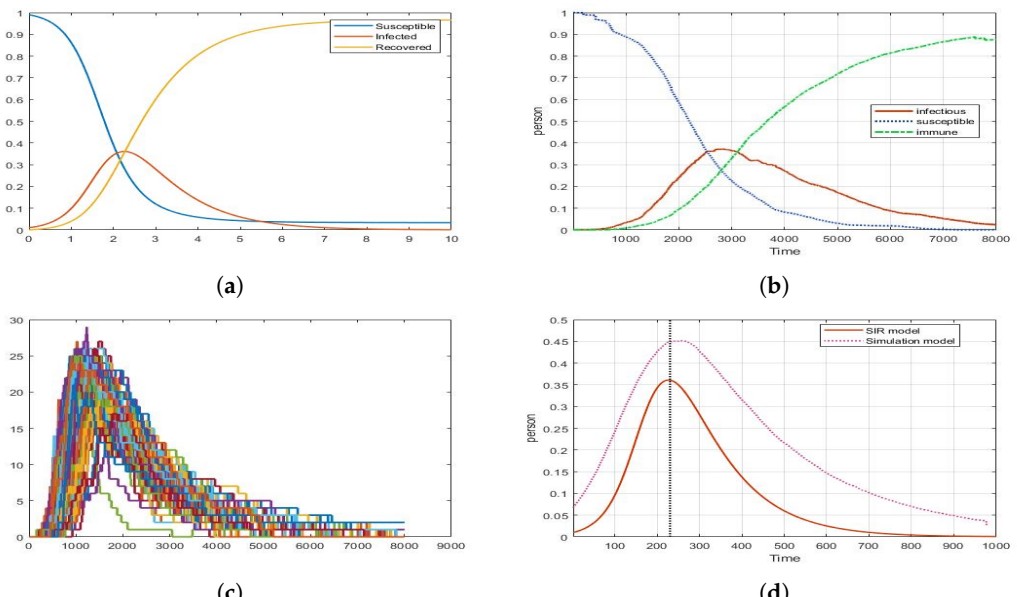

**Figure 5.** Graph of SIR model and simulation model when $R_0 = 3.4$: (**a**) SIR model ($R_0 = 3.4$); (**b**) simulation model ($R_0 = 3.4$); (**c**) dynamics of Infected in simulation model (100 reps); (**d**) dynamics of infected that matches the peak point through time scaling.

*4.2. First Experiment: No-Control*

In Figure 6, it shows the dynamics of infectious, susceptible, and recovered people in CP with different densities of GP. This result shows that the spread pattern is different depending on the density. Depending on the population density, the peak number, the time to reach the peak, and the spread period of the disease are different.

In Figure 6d, the CP in a densely populated city, the disease spreads quickly and causes great damage. On the other hand, small towns do less damage, and the spread time is late, but it lasts for a long time; therefore, even if the same $R_0$ value is applied, a different infectious trend line is expressed. Figure 6a shows that the number of infected people increases rapidly in the early stages of the disease in areas with high population density; however, as the population density decreased, a sharp increase in the number of infected is not observed. According to results, the proportion of the largest number of infected people is 68% in Seoul, 37% in the metro city, and 18% in the country, respectively. When the largest number of infected people occurs, Seoul has 1326 ticks, the metro city has 2765 ticks, and the country has 3890 ticks. Ticks are the same concept as time. Depending on the situation, ticks can be analyzed by comparing them with real time.

In Table 1, Infectious means the largest number of infected people. This value ranges from 0 to 1, and it is the average value for 100 replications. In addition, the value in parentheses is the variance of 100 data. The peak time is when the largest number of infections occur. In the third column (Surpass time 25%), we describe the time when the infected agent exceeds 25% in CP. The fourth column (Return time 25%) means when the number of infected people falls down 25% in CP. Hence, if the force required to maintain DEFCON is 75% of all troops, CP in Seoul loses its function after 755 ticks, and it takes 2158 ticks to return to 75%. So, if we assume that 8000 ticks correspond to 3 months, it means that the military is unable to perform its mission for a period of about a month. In addition, it takes 10 days to reach 25%, and in reality, it can be reached in a shorter time. There is a difference in the variance of the largest number of infected people by region. The larger the variance, the more varied the spread pattern per experiment. Conversely, if the variance is small, the experimental results are constant.

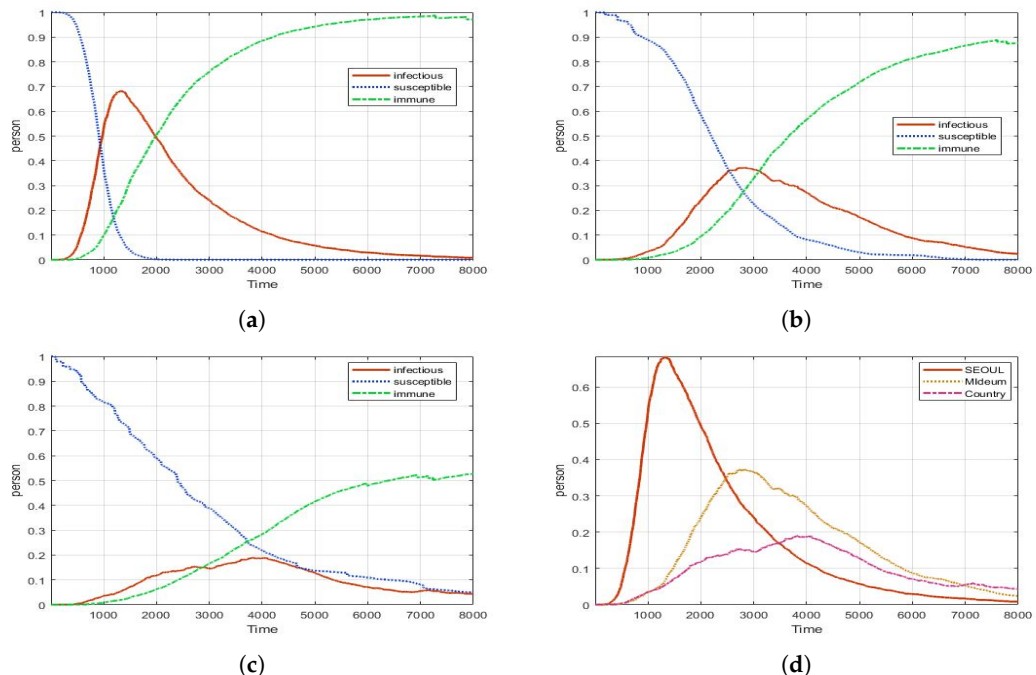

**Figure 6.** Experimental results for No-control (with different density of population). Blue line (- -): Susceptible, Red line (-): Infectious, Green line (-): Recovered. (**a**) Seoul (peak: 0.68, 1326 ticks); (**b**) Metro city (peak: 0.37, 2765 ticks); (**c**) Country (peak: 0.18, 3890 ticks); (**d**) Infectious for all areas.

**Table 1.** Summary of results (No-control).

|  | Infectious | Peak Time | Surpass Time (25%) | Return Time (25%) |
|---|---|---|---|---|
| Seoul | 0.68 (0.01) | 1326 ticks | 755 ticks | 2913 ticks |
| Metro | 0.37 (0.06) | 2765 ticks | 2060 ticks | 4153 ticks |
| Country | 0.18 (0.06) | 3890 ticks | - | - |

*4.3. Second Experiment: Social-Distance*

In Figure 7, the social distance effect is measured for the CP. In Seoul, when No-control was applied, the maximum number of infected people is 68%. In the same area, as a result of applying the Social-distance modifier, the maximum number of infected people is 43%. In other words, it is estimated that there is about 20% of block effect compared to No-control. In addition, it can be seen that the timing of the largest number of infections in each region is also slightly pushed backward. We conjecture that the number of contacts of agents reduced by Social-distance partially delayed the spread of the disease.

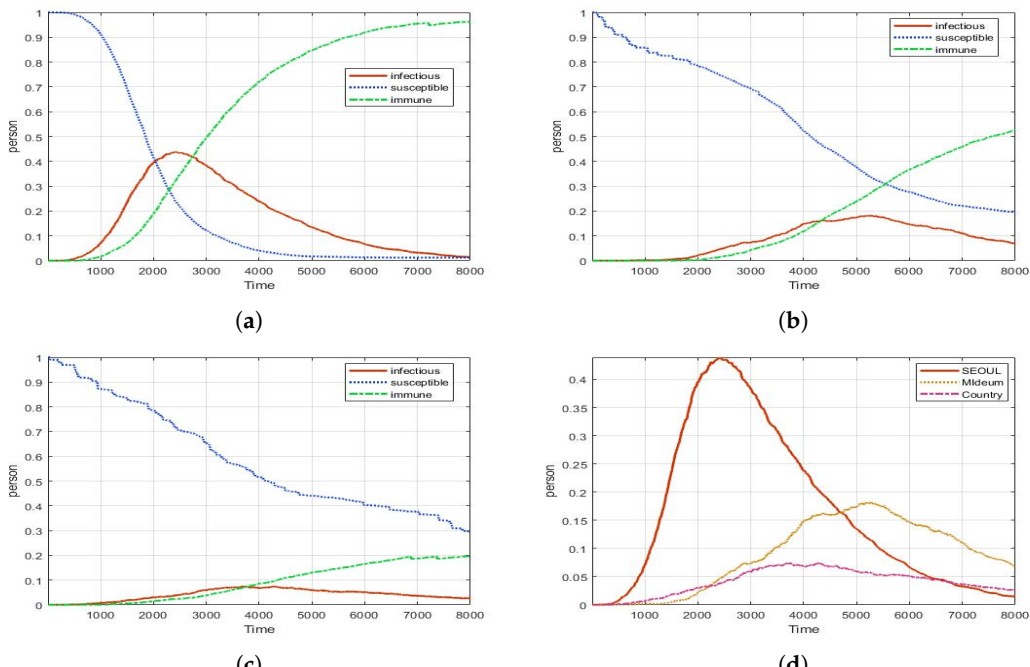

**Figure 7.** Experimental results for Social-distance (with different density of population): (**a**) Seoul (peak: 0.43, 2416 ticks); (**b**) Metro city (peak: 0.18, 5237 ticks); (**c**) Country (peak: 0.07, 3596 ticks); (**d**) Infectious for all areas.

### 4.4. Third Experiment: Screening

In Figure 8, when compared to No-control, Screening (fever check) reduces the number of infected by 30%. It maintains the safety of the CP quite stably. There is no rapid increase in the number of infected people. In addition, the majority of regions show an infected number close to or less than 25%. This number is important, because the absence of 25% of the total force is the maximum allowable value.

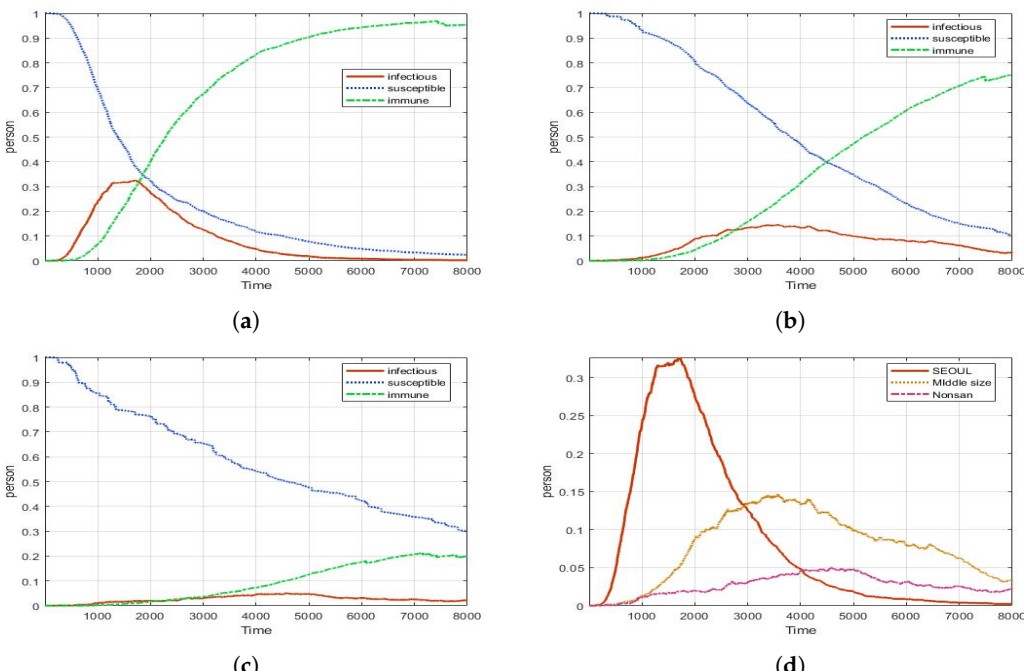

**Figure 8.** Experimental results for Screening (with different density of population): (**a**) Seoul (peak: 0.32, 1697 ticks); (**b**) Metro city (peak: 0.14, 3557 ticks); (**c**) Country (peak: 0.04, 4554 ticks); (**d**) Infectious for all areas.

### 4.5. Fourth Experiment: PCR Test

The PCR test shows the best prevention effect. In Figure 9, there are 40% fewer infections than in No-control, and most of the areas appear to be fairly stable. In this experiment, the control measure is applied to all agents exposed to the outside; however, it is not possible to conduct a test every time for all agents. In the real world, PCR tests are performed only when they are exposed to the outside for a long time or have contact with infected people; therefore, we reflect such property into our model. Such measure is called a Selective PCR test and is described in Section 4.6.

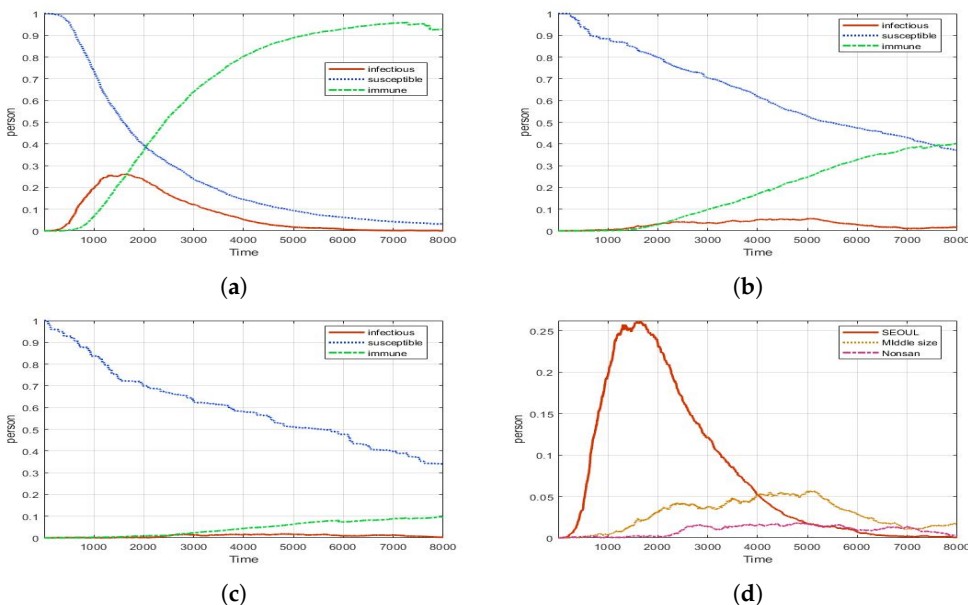

**Figure 9.** Experimental results for PCR test (with different density of population): (**a**) Seoul (peak: 0.26, 1668 ticks); (**b**) Metro city (peak: 0.05, 5070 ticks); (**c**) Country (peak: 0.01, 4886 ticks); (**d**) Infectious for all areas.

Table 2 shows the experimental results of Seoul by each control measure. The experimental results show that the effectiveness of all control measures varies.

**Table 2.** Summary of control measure effects (Seoul).

|  | Infectious | Peak Time | Surpass Time (25%) | Return Time (25%) |
|---|---|---|---|---|
| No-control | 0.68 (0.01) | 1326 ticks | 755 ticks | 2913 ticks |
| Social-distance | 0.43 (0.03) | 2416 ticks | 1529 ticks | 3872 ticks |
| Screening | 0.32 (0.02) | 1697 ticks | 1054 ticks | 2118 ticks |
| PCR test | 0.26 (0.02) | 1668 ticks | 1317 ticks | 1751 ticks |

The first one is from an Infectious point of view. The most effective control measure for reducing the number of infected is the PCR test. The PCR test shows the best effect in blocking the infected agent entering the CP. In addition, it can be confirmed that CP has the shortest period over which 25% or more of the infected are exceeded. The second point of view is the peak time. The earlier the peak point, we can see a rapid increase in the number of infections. From that point of view, Social-distance shows the characteristic of pushing the peak-point backward. However, it has been observed that the period in which more than 25% of infected people occur is the longest, and the spread of the disease is prolonged. So, if it is used alone, we should put up with pandemics for a long time. The third point is the period when the number of infected people exceeds 25%. The PCR test has most effective for maintaining below 25%. In addition, it can be seen that the Screening is also relatively stable. As a result, from the point of view of CP, PCR test is the best

control strategy; however, in reality, PCR test consumes a lot of time and cost. Therefore, similar to reality, it is necessary to selectively classify and apply agents. Hence, we tested the effectiveness of the Selective PCR test. In addition, we study the effect of combination control measures with CP in Section 4.6.

### 4.6. Fifth Experiment: Selective PCR Test and Combination Control Measures

Figure 10 is the result of the Selective PCR test. The Selective PCR test is a method performed on an agent exposed to the outside for a long time. For this experiment, the agents of CP are divided into two classes. The first class is an agent that has a short-term vacation. A total of 10% of all agents can freely enter and exit the CP. However, the agent cannot stay outside for a long time. The second class is the long-term vacation agent. They can stay outside for longer than short-term agents. This class accounts for 15% of all agents. As a result, 57% of the largest infections occurred at 1299 ticks. Surprisingly, this result is worse than the Social-distance with the lowest control effect. It seems likely that these results are in fact because only selected agents have the PCR test administered; therefore, Selective PCR test is not the best control measure in CP. Alternatively, Selective PCR test and Screening must be combined, because PCR test only applies to a few agents; however, Screening is applied every time to all agents. In other words, the combination of the two control measures can be said to be complementary to each other.

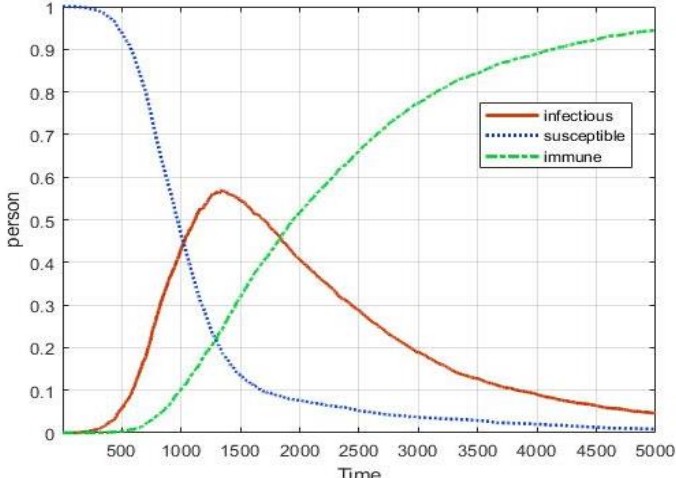

**Figure 10.** Selective PCR test (peak: 0.57, 1299 ticks). PCR test only for long-term agents.

In Figure 11, when Screening is combined, most show similar graphs regardless of short-term or long-term rate. Interestingly, if the Selective PCR test is performed at the same time, the CP shows stability regardless of the vacation rate. As a result, Screening must be prioritized in CP.

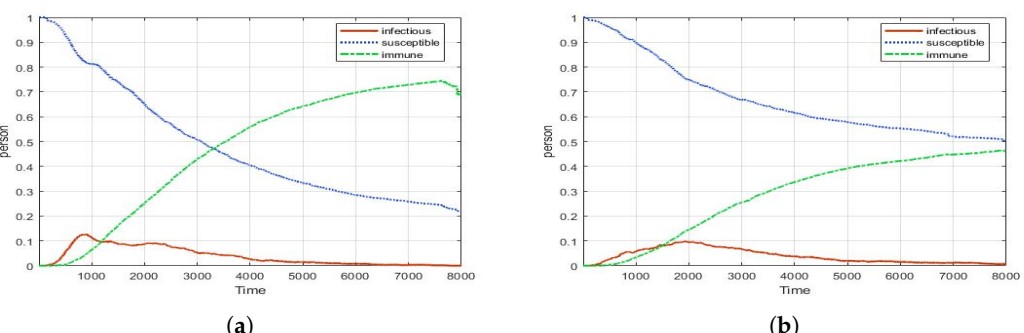

**Figure 11.** Experimental results for Screening + vacation rate + PCR test (long-term): (**a**) Long 15%, Short 10% (peak: 0.1); (**b**) Long 0%, Short 10% (peak: 0.1).

## 5. Results for Human Behavior

So far, the effect of control measures in CP has been described using the basic model. We can use this basic model for extended analysis. As an example of the application of the model, we conducted two additional experiment cases—Super-spreader and human mannerism.

### 5.1. First Experiment: Super-Spreader

A Super-spreader is a person who causes a lot more secondary infections than generally infected people [36]. They may be more active than normal people, and they may have a higher viral load [37]. When a Super-spreader moves in a limited space such as CP, the disease can spread rapidly due to the characteristic of being cut off from the GP. Therefore, it is important to block Super-spreaders from being active inside or coming in from the outside; however, it is difficult to predict, track, and manage the movement of Super-spreaders in advance; therefore, we conducted a check on the effectiveness of the control measures for blocking Super-spreaders. To reflect these characteristics in the model, we define a Super-spreader agent. They have a wide range of mobility, high speed, and high infection rate. In this model, these characteristics are set to be twice as large as the general agent. After that, we checked how much control measures can block Super-spreaders.

Figure 12a is a graph for the effect of a CP when Super-spreaders are active in GP. As the result, the maximum number of infections is 68%. Figure 12b shows a case where a Super-spreader is generated CP. At this time, up to 72% of infections occurred. This number shows that if Super-spreaders are move-in CP, it can rapidly increase the number of infected people; therefore, we check how many block Super-spreaders using the combination control measures.

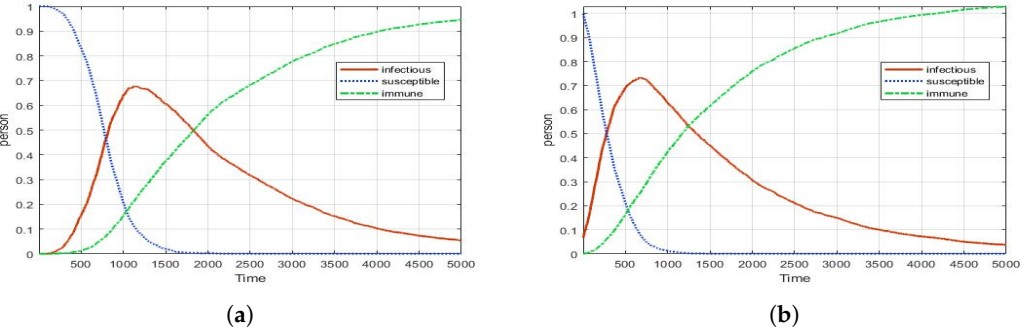

|     |     |
| :-: | :-: |
| (**a**) | (**b**) |

**Figure 12.** The effect of Super-spreaders whereabouts (No-control): (**a**) Super-spreader is in GP (peak: 0.68); (**b**) Super-spreader is in CP (peak: 0.72).

In Figure 13a, when both Screening and PCR test are applied to agents entering the CP, the number of infected people is less than 10% if there is no Super-spreader; however, if there is a Super-spreader, the maximum number of infections is as high as 44%, as shown in Figure 13b. When Super-spreaders are active, the combination control measures are about 30% effective. In addition, the increase in the number of infected is also made quite rapid; however, it can be seen that the disease is managed stably toward the latter part of the disease spread. The most striking finding of this experiment is that if an intentional enemy attack in CP (military) is made in the form of a Super-spreader, the CP (military) can easily lose its function for a certain period. Through this, we can think that these types of attacks can become non-conventional weapons in modern warfare.

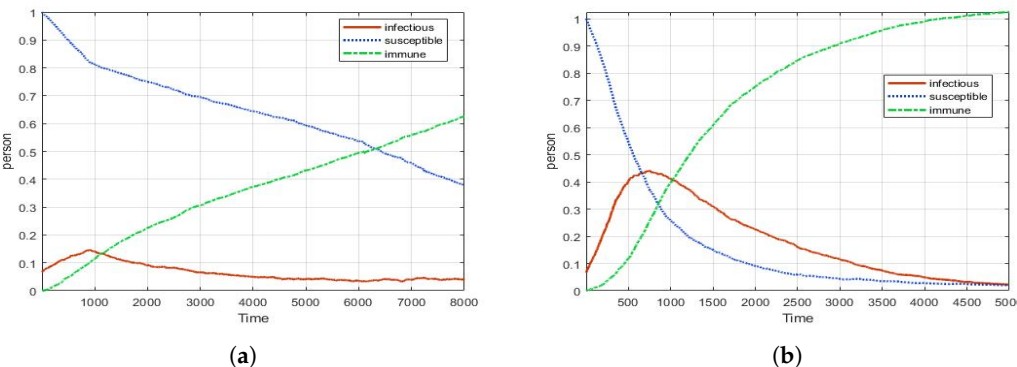

(**a**)                                                                                (**b**)

**Figure 13.** The effect of control measure (from Super-spreader). (**a**) No Super-spreaders (peak: 0.1);
(**b**) Super-spreaders present (peak: 0.44).

*5.2. Second Experiment: Mannerism*

Mannerism refers to the characteristic of a human being to become familiar with a certain situation [38]. When a disease spreads, people usually behave carefully in the early stages of the disease; therefore, most people follow personal quarantine rules well. Over time, some people forget the dangers of the disease. In this way, people keep repeating tension-relieving actions to prevent the spread of disease; therefore, we assume that human-behavior-related diseases would repeat tension and relaxation. In this model, the agent's speed is set to be less than 30% of its usual speed for human mannerism situations.

Figure 14 is a graph of the number of people infected with COVID-19 in a CP over the past year in Korea. Overall, these results imply that the 1st, 2nd, and 3rd waves in Korea affected the number of infected inside of the CP. In addition, it can be seen that the disease spreads by repeating it into both a stable and a rise phase. Further, we can confirm that the number of infected is increasing in the real world.

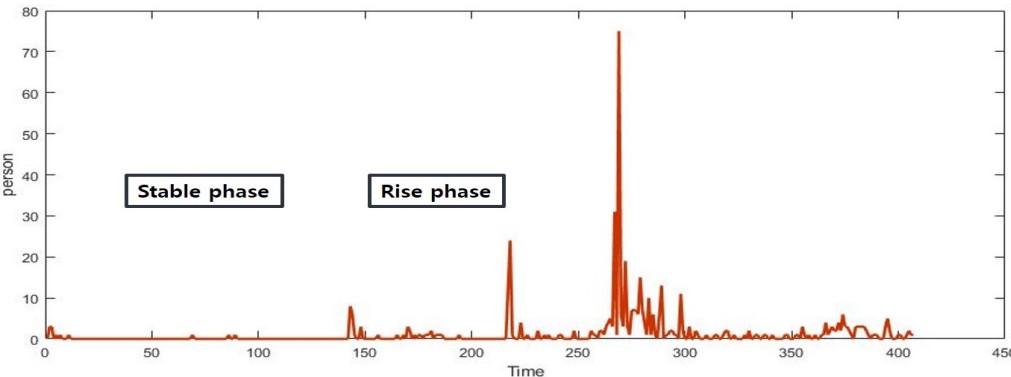

**Figure 14.** Real data on the number of infected person in CP (South Korea).

Figure 15 shows the effect of human mannerism in the model. As shown in Figure 15a, a graph is drawn skewed to the left by people's mannerisms. Interestingly, the shape of Figure 15a is strikingly similar to the depiction of trends in the number of infected in Figure 14. Figure 15b is the result of a combination control (Social-distance, Screening, PCR test). Even in human mannerism situations, the combination of control measures is effective. Consequently, the combination of control measures appears to effectively block the spread of CP under various conditions. Although the number of infected people rises slightly at the early stage of the disease spread, it remains stable overall; therefore, it is possible to secure the golden time of CP and keep DEFCON stable in groups such as the military.

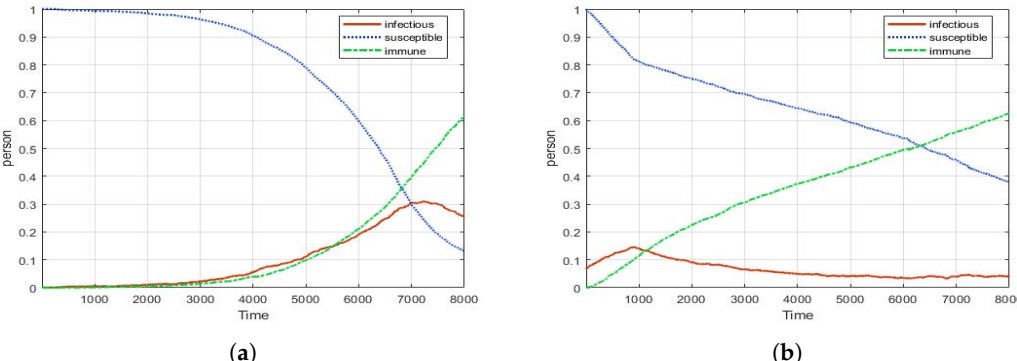

(**a**)                                                                                    (**b**)

**Figure 15.** The graph reflecting human mannerism: (**a**) No-control (peak: 0.3); (**b**) Combination control (peak: 0.14).

## 6. Conclusions

In this study, we implemented a disease spread simulation using ABM and investigated the effect of control measures for CP. We defined the properties of the CP and observed the effects of various control measures. The CP has different characteristics from the GP; therefore, timely and precise control is important. However, if disease information is limited, it is difficult to create models that test various control measures. Thus, in order to make a useful model from a small amount of information, we conducted an analysis of disease control measures using an agent-based model (ABM) that needs less computational or theoretical effort; therefore, this model helps to find the appropriate control measure in the CP using only a few pieces of information. In addition, this model specifically targets the military, but if only the characteristics of the group can be defined, the model can be changed in various ways.

According to the experimental results of the model, compared to No-control, Social distancing is 20% effective, Screening is 30%, and PCR test is 40% effective. The PCR test is the best. However, in the real world, PCR testing is difficult to consistently apply as a control measure; therefore, we found that the application of screening in conjunction with a selective PCR test is complementary for CP. Further, we have created various situations in the basic model. Through this, the danger of the Super-spreader is confirmed. Moreover, by reflecting on human mannerisms, the model is improved to be more realistic. In these special situations, combination control based on screening shows the result of effectively controlling the disease. In conclusion, the population that has intermittent contact with the outside world should undergo screening to maintain inside safety, and combination control, such as the selective PCR test is the most effective.

The proposed model can be flexibly adjusted for various situations. In addition, if the definition of a population is possible, it is possible to experiment with various hypotheses. In reality, difficult or time-consuming experiments can be performed quickly, and there is an economic advantage. Hence, the purpose of this experiment was to show the properties of the model and the many results it can provide. However, it is an experiment with reduced experimental space and cannot reflect the agent's diversity (occupation, age, gender, mutant virus, asymptomatic, etc) and life cycles such as death and birth. In addition, the effect of blocking the disease according to the inoculation rate of the vaccine has not been tested. In line with this, the model should be improved considering complex real world data such as density of population and the disease spreading in a complicated way. Furthermore, it is necessary to extend the model, such as modeling a real country. If this content is improved in the future, it will be possible to use it as a more realistic disease control model.

**Author Contributions:** Conceptualization, N.C. and Y.K.; methodology, Y.K.; validation, N.C. and Y.K.; formal analysis, Y.K.; investigation, Y.K.; resources, Y.K.; writing—original draft preparation, Y.K.; writing—review and editing, N.C.; visualization, Y.K.; supervision, N.C.; All authors have read and agreed to the published version of the manuscript.

**Funding:** This research received no external funding.

**Institutional Review Board Statement:** Not applicable.

**Informed Consent Statement:** Not applicable.

**Data Availability Statement:** Data sharing is not applicable to this article.

**Acknowledgments:** We would like to thank Jongcheol Kim who worked so diligently with us at Korea National Defense University.

**Conflicts of Interest:** The authors declare no conflict of interest.

## Appendix A

Rules for Control Measures:

[Social distance]
 for Agent $i_{\in A_{IN}}$
 **if** *Social distance applied* **then**
  *Agent i speed* $\leftarrow$ *Agent speed* $*$ $0.3(slowdown)$
 **else**
  *Agent speed*(*normal speed*)
 **end if**
[Screening]
 for Infected Agent $i_{\in A}$ reaches gate, generate $\delta \sim u[0,1]$
 **if** $\delta \leq 0.66$ **then**
  *No permission to CP.*
 **else**
  *Enter to CP.*
 **end if**
\* PCR-test : same to Screening rules but changes $0.66 \rightarrow 0.99$.

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
