# Peer review of "A Simulation Study on Spread of Disease and Control Measures in Closed Population Using ABM"

_computation, doi:10.3390/computation10010002_

Round 1
Reviewer 1 Report
The article presents an approach, unusual for civilian epidemiology and virology, to assessing the possibility and rate of spread of infectious diseases in closed groups of people and control measures for them by a computational simulation agent–based model. Evaluating this study as an expert in virology, I have to say that only a comprehensive analysis and comprehensive approaches may help humanity cope with infectious diseases, and, in particular, with the “plague of the 21st century” – COVID–19. From this point of view, this very interesting and timely study sounds like a proof of concept that can be extended from military to any other closed or semi–closed groups (polar stations, boarding schools, nursing homes, old–believers' congregations, etc.) or temporarily closed groups (for instance, aircraft passengers) and use for any infection disease control and prevention. The rarer is clear and well written and well–illustrated. The figures in particular are excellent and convey a lot of detailed information in a short space. However, some major modifications in the manuscript are required.
MAJOR COMMENTS
Point 1: The title of the article is great if throughout the article you are talking about ANY INFECTIOUS DISEASE; in this case, the title is perfect. But then don't start your Abstract with a description of COVID–19. If in the article you are talking about COVID–19, you must add "COVID–19" to the title. However, in the entire text, you only mention COVID–19 five times, including the Abstract. I propose to specify whether you are describing a situation (i) with any infectious disease, (ii) with COVID–19, or (iii) with COVID–19 as an example of one of the many infectious diseases. In the latter two cases, consideration should be given to changing the title of the article to make it more specific and consistent with the content.
Point 2: Through all the text it is not always clear what disease the authors write about – about ANY DISEASE or a SPECIFIC ONE (See also point 1). This is confusing and interferes with the perception of the text, which is in general written very clearly.
Point 3: Line 21–23. This sentence sounds very general and could be applied to any infectious disease. However, the next sentence (lines 23–24) sounds confusing (See also points 1-2). What particular disease are the authors talking about? Please, clarify these sentences.
Point 4: Line 59–65. Please, specify the exact disease you talking about. Is it COVID–19 or any infectious disease? (See also points 1-3).
MINOR COMMENTS
Point 5: Line 128. The authors said, “In Figure 2, we show a simple but useful disease spread model using ABM.” However, Figure 2 referred to reference [20] by other authors. From the ethical point of view, the sentence should be replaced with, for instance, “A simple but useful disease spread model using ABM is shown in Figure 2.”
Point 6: Figure 2. The title of the figure should be replaced with “Examples of disease models using ABM [20]).”
Point 7: Figure 3 should not be separated. Symbols presented in Figure 3 are legends for Figure 4. Thus, Figure 3 should be united with Figure 4.
Point 8: Figure 4. The images of symbols in the figure are very small and unreadable; they should be increased at least four times.
Author Response
Dear reviewer :
We thank the reviewer for your generous comments on the manuscript. We have edited the manuscript to address your concerns.
Below we address the reviewer comments and list of changes that we made to our manuscript according to your reports.
In addition, we believe that these modifications have strengthened the manuscript and hope that the revised manuscript is suitable for publication in MDPI Computation.
Point 1: The title of the article is great if throughout the article you are talking about ANY INFECTIOUS DISEASE; in this case, the title is perfect. But then don't start your Abstract with a description of COVID–19. If in the article you are talking about COVID–19, you must add "COVID–19" to the title. However, in the entire text, you only mention COVID–19 five times, including the Abstract. I propose to specify whether you are describing a situation (i) with any infectious disease, (ii) with COVID–19, or (iii) with COVID–19 as an example of one of the many infectious diseases. In the latter two cases, consideration should be given to changing the title of the article to make it more specific and consistent with the content.
Answer to Point 1 : We agree with your suggestion.
Our research presents a model and methodology that can be used for any infectious disease. Therefore, from your suggestion, (iii) with COVID-19 as an example of one of the many infectious diseases is the closest to our research direction. However, in order to show practical applicability of the research, we need to use one disease and corresponding data. Hence, we use information related to COVID-19, which has recently been intensively studied. In other words, this model is not focused on only COVID-19.
So, the modified part is as follows :
1) line 1 : A disease such as COVID-19 -> An infectious disease
2) line 10-11 : COVID-19 is an infectious disease that first emerged in Wuhan, China in 2019.
-> The spread of infectious diseases has a great impact on human society.
COVID-19 is the best representative disease that is currently impacting human society.
It is an infectious disease that first occurred in Wuhan, China in 2019.
In this way, the abstract and introduction have been revised to reduce confusion among readers.
Point 2 : Through all the text it is not always clear what disease the authors write about – about ANY DISEASE or a SPECIFIC ONE (See also point 1). This is confusing and interferes with the perception of the text, which is in general written very clearly.
Answer to Point 2 : Your suggestion is correct. In connection with point 1, there are parts that confuse the reader as we use COVID-19 data and a term. We modified the introduction to correct the context. Through these revisions, we believe that our manuscript is clearly present our research direction with related to scope of disease.
So, the modified part is as follows :
1) line 59 - 62 : None -> Before going into further details, it is required to notice that we use
COVID-19 data for empirical research. However, this model is not focused
on only COVID-19. The disease spread model in this study can be extended
to any infectious disease.
Point 3 : Line 21–23. This sentence sounds very general and could be applied to any infectious disease. However, the next sentence (lines 23–24) sounds confusing (See also points 1-2). What particular disease are the authors talking about? Please, clarify these sentences.
Answer to Point 3 : We agree with your suggestion. The Cheonghae Unit is an event caused by
COVID-19. So, we expressed the content clearly.
1) line 24-26 : For instance, in July 2021, the large-scale spread of disease occurred at the
Cheonghae naval unit (i.e., the Republic of Korea Navy Somali Sea Escort Task
Group). -> For instance, in July 2021, a large-scale spread of COVID-19
occurred at the Cheonghae Naval unit (it is the Republic of Korea Navy Somali Sea
Escort Task Group).
Point 4 : Line 59–65. Please, specify the exact disease you talking about. Is it COVID–19 or any infectious disease? (See also points 1-3).
Answer to Point 4 : We agreed. Our aim of Section 2(Literature Review) is to give better understanding about disease spread model for ANY infectious disease. In connection with answer for point 1 and 2, we modified corresponding content clearly.
1) line 64-69 : There are various models that explain the mechanisms of disease. Many researchers have sought to determine the characteristics and scale of diseases using disease models. Results using such models can be used to establish policies against disease. As diseases continue to evolve and change, researchers attempt to construct robust and accurate models. In general, disease models can be divided into mathematical models and simulation models. ->
There are various models that explain the mechanisms of infectious disease. Many researchers study to determine the characteristics and scale of infectious diseases using disease spread models. Results using such models can be used to establish policies against disease. As diseases continue to evolve and change, researchers attempt to construct robust and accurate models. In general, disease spread models can be divided into mathematical models and simulation models.
Point 5 : Line 128. The authors said, “In Figure 2, we show a simple but useful disease spread model using ABM.” However, Figure 2 referred to reference [20] by other authors. From the ethical point of view, the sentence should be replaced with, for instance, “A simple but useful disease spread model using ABM is shown in Figure 2.”
Answer to Point 5 : Thank you for your exact and kind correction. Works described in Figure 2 are not obviously our work and sentence should be modified as you suggested.
The corrections are as follows :
1) line 133 : In Figure 2, we show a simple but useful disease spread model using ABM. -> A simple but useful disease spread model using ABM is shown in Figure 2.
Point 6 : Figure 2. The title of the figure should be replaced with “Examples of disease models using ABM [20]).”
Answer to Point 6 : We agree to modify the title. Your recommendation is more accurate.
The corrections are as follows : Figure 2. Examples of disease models using ABM (Net-logo library (Wilensky, 1999) [20]). ->Figure 2. Examples of disease models using ABM [20]).
Point 7 : Figure 3 should not be separated. Symbols presented in Figure 3 are legends for Figure 4. Thus, Figure 3 should be united with Figure 4.
Answer to Point 7 :
As you suggested, we disposed two figures in same page and it makes readability better. However, we did not combine two figures together (use one figure as legend). We have two reasons.
First, each figure has subtitles those have their roles. It turned out that combining two figures leads to complicated description of figure title. Second, we wanted to explain agents and environment of the model separately. If agent figure is inserted into environment figure, reader may not pay particular attention to agents which is indeed important element of the model. We hope that you are satisfied with our suggestion.
Point 8 : Figure 4. The images of symbols in the figure are very small and unreadable; they should be increased at least four times.
Answer to Point 8 : Thank you for your good concern. As you said, we increased four times the size of the symbol(Figure 4).
Thanks for the good review, please refer to the revised thesis together

Reviewer 2 Report
The paper represents interesting simulations using data on the spread of Covid-19 in South Korea. The main positive contributions of the paper are the numerical modeling, the description of the simulation model, and the problem statement. However, there are comments:
- The literature review should be extended. Now it is not clear the structure of section 2. On the one hand, it looks like the unfinished literature review, on the other hand, the section represents some mathematical frameworks and simulation models. It is not very clear from sections 2.1. and 2.2. whether the authors are presenting their results or the results of previous studies.
- It would be recommended to add a formal representation of the studied model, not only a schematic representation.
- The authors discussed asymptomatic cases in section 4.2, but this group does not include in the model. It can be an interesting extension of the model.
- The paper contains many good simulations, unfortunately, there is no information on how the control measures are included in the model. Now readers can see only the results. It can be interesting to follow the formal algorithms which were used to receive the final results of simulations.
Author Response
Dear reviewer :
We thank the reviewer for your generous comments on the manuscript. We have edited the manuscript to address your concerns.
Below we address the reviewer comments and list of changes that we made to our manuscript according to your reports.
In addition, we believe that these modifications have strengthened the manuscript and hope that the revised manuscript is suitable for publication in MDPI Computation.
1. The literature review should be extended. Now it is not clear the structure of section 2. On the one hand, it looks like the unfinished literature review, on the other hand, the section represents some mathematical frameworks and simulation models. It is not very clear from sections 2.1. and 2.2. whether the authors are presenting their results or the results of previous studies.
Answer : First of all, all researches mentioned in Section 2.1 and 2.2 are not our findings. Our aim of Section 2 is provide better understanding about two representative disease modeling methodologies; SIR model and ABM simulation model. We admit that some explanations are too detail so that reader is confused. However, we believed that it was necessary for finding where our work is positioned and what is contribution of our work. We hope that you are satisfied with our answer.
2. It would be recommended to add a formal representation of the studied model, not only a schematic representation.
Answer : Thank you for your good suggestions. In general, mathematical problem(optimization) have formal representation including objective function and constraints. As far as we know, however, ABM does not have such a formal expression. For examples, [Wilensky, Uri, and William Rand. An introduction to agent-based modeling: modeling natural, social, and engineered complex systems with NetLogo. Mit Press, 2015.] introduce various ABM models (i.e., El Farol model, Spread of Disease Model, The Fire Model, etc) with descriptions on agents, environments, and parameters necessary to design a model. Some rules are expressed in language or using equations. Recent ABM works, for instance [Ferrare, F., et al. "Urban Air Mobility (UAM): A Model Proposal based on Agents using Netlogo." Proceedings of the 11th International Conference on Simulation and Modeling Methodologies, Technologies and Applications. 2021.], also follow same structure. And we believe that the structure of our research is not much different from other ABM researches. Nevertheless, we understand and agree with your concerns and we add rules for Control Measures (in algorithm form) at Appendix section (this is also related to answer 4)
3. The authors discussed asymptomatic cases in section 4.2, but this group does not include in the model. It can be an interesting extension of the model.
Answer : As per your suggestion, the asymptomatic case can be reflected by adding another agent and it will be a great idea. However, in our study, we did not create asymptomatic agents. It is because we design our model referred to the SIR model. SIR model explains complicated disease spreading phenomena by only using S, I, and R group and so is ours. However, our model somehow reflect asymptomatic properties fortunately. That is, the asymptomatic property is reflected through probability of control measures. For example, in the case of screening, there is a 66% probability that the infected agent will be blocked from entering the CP. In other words, there is a 33% chance that an infected agent can enter the CP. In this case, 33% is a reflection of the proportion of asymptomatic infections. Reflecting asymptomatic cases to agents seems like a great idea. So, we plan to create a model that reflects various agents in further studies.(Asymptomatic cases are reflected in the further study.(line 440)
4. The paper contains many good simulations, unfortunately, there is no information on how the control measures are included in the model. Now readers can see only the results. It can be interesting to follow the formal algorithms which were used to receive the final results of simulations.
Answer : We agree with your suggestion. Although we described the rules for agents, environments, and parameters in detail, it seems that the explanation of control measures is insufficient.
So, we include formal expression detailed rules on how to control measure are applied of each control measure in Appendix A. The reason why we do not include such part into main body of the research is that it may be enough to explain such simple rules in language.
Thanks for the good review, please refer to the revised thesis together

Reviewer 3 Report
This manuscript reports results of the agent-based simulations of epidemic spread under free conditions and various spread-preventing measures. The simulation scheme is well-motivated, the results of simulations behave reasonably from the point of view of mathematical epidemiology. The discussion, how different measures affect an outbreak under various conditions provides potentially useful information.
At the same time, there are some comments aimed to improve the clarity of the research:
1) Section "3.3. The agent behavior": does "Random number" mean a random number taken from a uniform flat distribution? Eqs. (6) and (7) allow making such a conclusion but it would be better to state the type of random number distributions explicitly.
2) Fig. 5 compares epidemic curves obtained in the deterministic and stochastic models. The simulated curve in Fig. 5(b) is obtained by averaging over the ensemble. Since the process is non-linear and non-Gaussian, maybe it makes sense to plot a distribution, which may be long-tailed and use median instead of mean?
3) How to correspond ticks of simulation to dimensional time, e.g. days?
4) I can note the article Postnikov, E. B. (2020). Estimation of COVID-19 dynamics “on a back-of-envelope”: Does the simplest SIR model provide quantitative parameters and predictions?. Chaos, Solitons & Fractals, 135, 109841, which provides a more detailed analysis of the applicability of the SIR model during the initial phase of COVID-19 pandemics, and, additionally, mentions deviations of the epidemic curve for South Korea from an ideal SIR system's solution that can be associated with the anti-spread measures discussed in the present manuscript based on the situation in the mentioned country.
Author Response
Dear reviewer :
We thank the reviewer for your generous comments on the manuscript. We have edited the manuscript to address your concerns.
Below we address the reviewer comments and list of changes that we made to our manuscript according to your reports.
In addition, we believe that these modifications have strengthened the manuscript and hope that the revised manuscript is suitable for publication in MDPI Computation.
1) Section "3.3. The agent behavior": does "Random number" mean a random number taken from a uniform flat distribution? Eqs. (6) and (7) allow making such a conclusion but it would be better to state the type of random number distributions explicitly.
Answer : Thank you so much for your good suggestion. We used the NetLogo program and we use random number generator of it, called the 'random'. The ‘random’ generates number that follows uniform distribution. That is, the program is structured in the following form: ceil(Uniform(0,1)* (N-1)), * N is User's input number and ceil means ceiling function. For example, in NetLogo program, random will give us numbers within the range from 0 to N - 1, instead of 1 to N. Therefore, every time we run ‘random 5’, there is an equal likelihood of getting 0, 1, 2, 3, or 4.
The corrections are as follows :
line 215 : none -> Random number follows uniform distribution in [0,1].
equation 6 -> Random number δ ∈ [0, 1] ≤ Transmission rate
equation 7 -> Random number δ ∈ [0, 1] ≤ (1/recovery day)
2) Fig. 5 compares epidemic curves obtained in the deterministic and stochastic models. The simulated curve in Fig. 5(b) is obtained by averaging over the ensemble. Since the process is non-linear and non-Gaussian, maybe it makes sense to plot a distribution, which may be long-tailed and use median instead of mean?
Answer : Thank you for your interesting suggestion. This process is non-linear and non-Gaussian. So, we conducted additional experiments to compare the difference between the median and mean.
Please refer to graph that shows the dynamics of infectious(No-control) where red line is mean of observations and blue line is median of observations.
The graph looks as follow:
https://drive.google.com/file/d/1XoVY4S1NdlulcTJBbo4Z8PZ1TbZsyBN7/view?usp=sharing
We conjecture that there are no significant differences between the two lines. We hope that you are satisfied with our answer.
3) How to correspond ticks of simulation to dimensional time, e.g. days?
Answer : In this study, 1 tick is amount to 1/80 day. (that means 80 ticks is a day.)
However, we also had concerns in conversion from tick to real-time. For this reason, we have studied lots of researches and it tuned out that there is no clear rule to define ticks.
Therefore, we decided to make an assumption on ticks empirically.
An explanation about ticks and time is in line 286.
4) I can note the article Postnikov, E. B. (2020). Estimation of COVID-19 dynamics “on a back-of-envelope”: Does the simplest SIR model provide quantitative parameters and predictions?. Chaos, Solitons & Fractals, 135, 109841, which provides a more detailed analysis of the applicability of the SIR model during the initial phase of COVID-19 pandemics, and, additionally, mentions deviations of the epidemic curve for South Korea from an ideal SIR system's solution that can be associated with the anti-spread measures discussed in the present manuscript based on the situation in the mentioned country.
Answer : Thank you very much for your valuable recommendation. The methodology presented in this article gave us good insight. Accordingly, We decided to include this research to Section 2(Literature review) line 97.
The corrections are as follows :
None -> These mathematical models are still useful for predicting disease spread. In addition, by
improving the accuracy of the SIR model, it gives good insight for policy evaluation.
Thanks for the good review, please refer to the revised thesis together.

Reviewer 4 Report
Obviously this paper should be published. First it uses ABM as an approach. Equation based models to understand epidemics are basically invalid. The difference between a contained infection and an epidemic is a phase transition. Equations are invalid instruments to analyze that kind of phase transition, which in this case is a percolation. Very slowly epidemiologists are learning that, which has been known to physicists for more than 30 years… ABMs are the only reasonable models to understand epidemics, even if they are more cumbersome than equations.
Now there are ABMs and ABMs. I would not argue that the ABM used in this paper could not be improve. It can and I am sure the authors in their next paper will do it. In this case the ABM was used helps them compare the situation of closed population (CP) and general population (GP). They found some structure. But if they are given the resources to be more ambitious and try to model realistically the spread of diseases in complex situations like real countries where the density of population is not uniform (never is) and where as a result the disease spreads in a complicated way, they are bound to find insights on how diseases like Covid spread, for the benefit of all of us.
So my diagnosis on that paper is that it is a good start, your work should be published as is, but “keep the good work”.
Author Response
Dear reviewer :
We thank the reviewer for your generous comments on the manuscript. We have edited the manuscript to address your concerns.
Below we address the reviewer comments and list of changes that we made to our manuscript according to your reports.
In addition, we believe that these modifications have strengthened the manuscript and hope that the revised manuscript is suitable for publication in MDPI Computation.
1) Obviously this paper should be published. First it uses ABM as an approach. Equation based models to understand epidemics are basically invalid. The difference between a contained infection and an epidemic is a phase transition. Equations are invalid instruments to analyze that kind of phase transition, which in this case is a percolation. Very slowly epidemiologists are learning that, which has been known to physicists for more than 30 years… ABMs are the only reasonable models to understand epidemics, even if they are more cumbersome than equations.
Now there are ABMs and ABMs. I would not argue that the ABM used in this paper could not be improve. It can and I am sure the authors in their next paper will do it. In this case the ABM was used helps them compare the situation of closed population (CP) and general population (GP). They found some structure. But if they are given the resources to be more ambitious and try to model realistically the spread of diseases in complex situations like real countries where the density of population is not uniform (never is) and where as a result the disease spreads in a complicated way, they are bound to find insights on how diseases like Covid spread, for the benefit of all of us.
So my diagnosis on that paper is that it is a good start, your work should be published as is, but “keep the good work”.
1) Thank you so much for your positive review. As you said, we created this model to provide insight on disease spread. In order to prepare for Post-COVID era, we need to improve our model that can be applied to various environments and can be tested in various ways. In the future, we plan to improve models including, but not limited to the herd immunity effect of vaccines and antiviral effect.
We include your valuable suggestions on further study.
Thank you again for your kind directions.
The corrections are as follows :
line 444 : None -> In line with this, the model should be improved considering complex real world data such as density of population, disease spreading in a complicated way. Furthermore, it is necessary to extend the model, such as modeling a real country. If this content is improved in the future, it will be possible to use it as a more realistic disease control model.
Thanks for the good review, please refer to the revised thesis together.

Round 2
Reviewer 2 Report
Thank you very much for revision.
Please find small comment into attached file.

Author Response
Please refer to the attached file for corrections.
Edited contents are marked in blue and deleted contents are marked in red.

This manuscript is a resubmission of an earlier submission. The following is a list of the peer review reports and author responses from that submission.